# Identification of a Novel IncHI1B Plasmid in MDR *Klebsiella pneumoniae* 200 from Swine in China

**DOI:** 10.3390/antibiotics11091225

**Published:** 2022-09-09

**Authors:** Huixian Liang, Xinhui Li, He Yan

**Affiliations:** 1School of Food Science and Engineering, South China University of Technology, Guangzhou 510641, China; 2Department of Microbiology, University of Wisconsin-La Crosse, La Crosse, WI 54601, USA; 3Guangdong Province Key Laboratory for Green Processing of Natural Products and Product Safety, Guangzhou 510641, China

**Keywords:** *Klebsiella pneumoniae*, plasmid, antibiotic resistance genes, whole genome sequencing

## Abstract

Multidrug-resistant (MDR) *Klebsiella pneumoniae* poses a seriously threat to public health. The aim of this study was to better understand the genetic structure of its plasmids and chromosomes. The whole-genome sequence of *K. pneumoniae* 200 isolated from the liver of a swine with diarrhea in China was determined using PacBio RS II and Illumina MiSeq sequencing. The complete sequences of the chromosomal DNA and the plasmids were analyzed for the presence of resistance genes. The phylogenetic trees revealed that *K. pneumoniae* 200 displayed the closest relationship to a human-associated *K. pneumoniae* strain from Thailand. *K. pneumoniae* 200 contained two plasmids, pYhe2001 and pYhe2002, belonging to the incompatibility groups IncH-HI1B and IncF-FIA. The plasmid pYhe2001 was a novel plasmid containing four types of heavy metal resistance genes and a novel Tn*6897* transposon flanked by two copies of IS*26* at both ends. Mixed plasmids could be transferred from *K. pneumoniae* 200 to *Escherichia coli* DH5α through transformation together. This study reported the first time a novel plasmid pYhe2001 from swine origin *K. pneumoniae* 200, suggesting that the plasmids may act as reservoirs for various antimicrobial resistance genes and transport multiple resistance genes in *K. pneumoniae* of both animal and human origin.

## 1. Introduction

*Klebsiella pneumoniae*, a gram-negative bacterium, is an opportunistic pathogen that can not only colonize the gastrointestinal tracts of healthy humans and animals [1] but also causes invasive diseases in swine [2]. With an increasing number of antibiotics used in livestock farms, *K. pneumoniae* has shown the ability to resist multiple antibiotics, even some last-resort ones (e.g., carbapenem and colistin) [3,4], commonly through acquiring preexisting resistance and virulence genes via plasmids and transposable elements [5].

Spread of resistance and virulence genes in *K. pneumoniae* can result from the conjugation or mobilization functions of plasmids. For example, hypervirulent *K. pneumoniae* from the sputum of an older male patient was found to harbor a *bla*_CTX-M-24_ carrying an IncFII-type plasmid and a pK2044-like virulent plasmid isolated from blood of a patient with liver abscess and meningitis, due to plasmid conjugation [6]. Furthermore, other transposable elements, such as transposons, play an important role in the spread of resistance genes [5]. For example, the *bla*_KPC_ gene has been identified within a Tn*3*-family transposon, Tn*4401*, in clinical *K. pneumoniae* [7]. Moreover, antibiotic resistance genes could transfer between plasmids and chromosomes through transposable elements. For example, a multidrug-resistant *K. pneumoniae* CN1 from a patient was found to carry both *bla*_CTX-M-15_ and *bla*_KPC-2_ genes in the chromosome with the *bla*_CTX-M-15_ gene linked to an insertion sequence, IS*Ecp1*, and the *bla*_KPC-2_ gene in Tn*4401a* [8], while IS*Ecp1*-*bla*_CTX-M-15_ and Tn*4401a* could be also found in plasmid pNY9_3 from *K. pneumoniae* NY9 and plasmid pCR14_3 from *K. pneumoniae* CR14, respectively.

Whole-genome sequencing (WGS) is rapidly becoming a powerful tool for identifying pathogenic features in *K. pneumoniae*, particularly for understanding the relative evolution of strains and the potential course of spread of mobile elements [9]. Through WGS, a previous study found that the megaplasmid in the *K. pneumoniae* isolate was a result of cointegration of an IncA/C2-type plasmid harboring *bla*_OXA-427_ with an IncF1b-type plasmid [10].

So far, most studies have focused on clinical isolates by detecting emerging resistance genes, such as carbapenemase genes or mobile colistin resistance genes in *K. pneumoniae*. However, researchers have rarely aimed to identify genomic elements of isolates from animals in plasmids or in the chromosome of nonhypervirulence *K. pneumoniae*. In this study, we identified an MDR *K. pneumoniae* 200 isolate from swine liver with a novel IncHI1B plasmid and an IncFIA plasmid with the *bla*_CTX-M-27_ gene, and investigated the mobile elements of *K. pneumoniae* to expand the understanding of MDR *K. pneumoniae*.

## 2. Results and Discussion

### 2.1. Characterization of K. pneumoniae 200

K. pneumoniae 200 exhibited resistance to β-lactams, aminoglycosides, florfenicol, streptomycin, quinolones, and sulfonamides (Appendix A).The complete genome of *K. pneumoniae* 200 contained a circular 5,257,665 bp chromosome with a total of 5106 ORFs and G + C content of 58.78% and two plasmids: pYhe2001 (213,254 bp) with a total of 290 ORFs and GC content of 50.10% and pYhe2002 (75,320 bp) with a total of 107 ORFs and GC content of 51.65%. Multilocus sequence typing analysis showed that *K. pneumoniae* 200 belonged to sequence type 37 (ST37) [11,12].

### 2.2. Characterization of Chromosome of K. pneumoniae 200

There were 71 resistance genes and 65 virulence genes in the chromosome. Resistance genes identified in the chromosome included genes for β-lactamases, efflux pumps, quinolone, fosfomycin, multidrug resistance genes (mdtK, tolC, and acrAB), the multiple antibiotic resistance operon (marRA), and resistance to heavy metals (copper, silver, and arsenic). In *K. pneumoniae*, the best-characterized efflux pump systems include acrAB and mdtK complexes [13]. Furthermore, the multidrug efflux pump system (acrAB-tolC) in *K. pneumoniae* isolates was related to resistance to quinolones, tetracyclines, tigecycline, and β-lactams leading to MDR [14]. Overall, the combination of the acrAB-tolC and mdtK genes was strongly associated with the MDR function of K. pneumoniae. The more detailed features of the *K. pneumoniae* 200 genome are listed in Appendix A.

Genome analyses revealed that T6SS components of *K. pneumoniae* 200 were highly similar (99% nucleotide sequence identity) to that of *K. pneumoniae* ZYST1 [15] (accession number CP031613): A main cluster and an auxiliary cluster were similar to those found in K. pneumoniae HS11286 and *K. pneumoniae* ZYST1 (Appendix A). Compared with strains HS11286 and ZYST1, the main T6SS cluster in *K. pneumoniae* 200 was conserved except for a portion of the effector and immunity genes and the paaR region. Furthermore, IS4 was found to be inserted upstream of the vipA gene in the main T6SS cluster. The auxiliary T6SS cluster also showed a conserved region and variable effector and immunity genes compared to strains HS11286 and ZYST1. BLASTN analyses revealed that there were protein homologs of the regulatory proteins TfoX and QstR as well as the competence proteins ComEA and ComEC in K. pneumoniae 200, suggesting the potential ability of uptaking genes and recombination consequences via the combination of natural competence and target cell killing mediated by T6SS [15]. The phylogenetic trees of all ST37 *K. pneumoniae* strains in GenBank revealed that the isolates from the same countries clustered on the same branches and were closely related (Figure 1). *K. pneumoniae* 200 branched from *K. pneumoniae* F10 (AN) from China and displayed the closest relationship to human-associated *K. pneumoniae* 4300STDY6470454 (UFHS01) from Thailand, suggesting that they may share a common origin.

### 2.3. Characterization of Plasmids Carried by K. pneumoniae 200

*K. pneumoniae* 200 contained two plasmids, pYhe2001 and pYhe2002. The plasmid pYhe2002 belonged to the IncF type and contained IncFIA replicon: FIA. The IncF-type plasmids were narrow-host range plasmids that are frequently identified among *Enterobacteriaceae* strains, especially *K. pneumoniae* [16]. BLASTN comparison revealed that the backbone of pYhe2002 was highly similar (99% nucleotide sequence identity with a query coverage of 98%) to that of the plasmid p19110124-2 (accession number NZ_CP064179.1), which originated from a *K. pneumoniae* strain isolated from an anal swab of swine in China. Comparison of the plasmid sequence with the plasmid records in PLSDB [15,17,18] using mash dist with maximal *p* value and distance thresholds set to 0.1, showed a result with 681 hits (Appendix A). The plasmids included in the hits were mostly from *K. pneumoniae* and the remaining plasmids were from other *Enterobacteriaceae* species (e.g., *Escherichia coli*, *Salmonella enterica*, and *Citrobacter freundii*). These plasmids were in isolates from humans (e.g., clinical patients), food (e.g., pork and milk), the environment (e.g., rivers and air), and animals (e.g., ducks, rabbits, chickens, and swine). The plasmid pYhe2002 contained antibiotic resistance genes *tetA*, *floR*, *bla*_CTX-M-27_, *sul1*, *qnrB2*, Δ*qacE*, *aadA16*, *dfrA27*, *arr-6*, and *aac(6**’)-Ib-cr* (Appendix A). The IncF transmissible novel plasmid with *bla*_CTX-M-27_ was identified as *E**. coli* isolated from swine in China [19]. In a previous study, of the 24 IncF-type plasmids analyzed, 22 *bla*_CTX-M-27_-carrying plasmids were identified in *E. coli* [20], suggesting that the backbone of the IncF plasmid may be a major transport for dissemination of the *bla*_CTX-M-27_ gene. The expression of CTX-M β-lactamase genes were found commonly in *K. pneumoniae* strains all over the world. Many kinds of CTX-M allele were found, such as *bla*_CTX-M-55_ [21], *bla*_CTX-M-63_ [22], and *bla*_CTX-M-15_ [23,24]. The predominant ESBL allele *bla*_CTX-M-15_ was commonly detected in IncF plasmids from *K. pneumoniae* [23,24]. From the results of PLSDB, among the 681 hits, gene *bla*_CTX-M-27_ could be found in plasmids carried by *K. pneumoniae* from different sources but not liver source (Appendix A). This is the first report that an IncFIA plasmid with the *bla*_CTX-M-27_ gene from swine liver *K. pneumoniae* isolate. The plasmid pYhe2002 carried a *sul1*-type class 1 integron that has a 5’CS and four cassettes, *aadA16*-*dfrA27*-*arr-6-aac(6’)-Ib-cr*, and 3’CS. The *sul1*-type class 1 integron was flanked by a gene for IS*6* family transposase in the upstream and a gene encoding DUF4440 domain-containing protein, a gene encoding peptide ABC transport, and an IS*91* family transposase gene in the downstream.

The plasmid pYhe2001 belonged to the IncH type and contains IncH replicon: HI1B. For comparison of the plasmid with plasmid records in PLSDB using mash dist with maximal *p* value and distance thresholds set to 0.1, the search resulted in 915 hits (Appendix A). The plasmids included in the hits were mostly from *K. pneumoniae*, and the remaining plasmids were from other Enterobacteriaceae species (e.g., *Escherichia coli*, *Salmonella enterica*, and *Citrobacter freundii*). These plasmids were collected from humans (e.g., clinical patients), food (e.g., pork and milk), the environment (e.g., rivers and air), and animals (e.g., chickens and swine). Previous studies showed that the plasmids from *K. pneumoniae* shared more than 85% query coverage with plasmid records [15,25]. Combining BLASTN comparison with PLSDB, the backbone of these plasmids is currently lower than 80% query coverage of pYhe2001, suggesting that pYhe2001 is a novel plasmid. Figure 2 shows the four plasmids with the highest similarity compared with pYhe2001.

The plasmid pYhe2001 showed query coverage of 78% in the plasmid pVNCKp115 (accession number LC549807.1), a plasmid from a *K. pneumoniae* isolate in Vietnam. Compared with pVNCKp115, four types of heavy metal resistance genes and a novel transposon, Tn6897, were specific to the plasmid pYhe2001. The plasmid pYhe2001 consisted of partition (parB/parA), transfer (traI) functions, a 26.5 kb MDR region as well as tellurium resistance-associated, mercury resistance-associated, silver resistance-associated [26,27], and copper resistance-associated genes. The MDR region comprised two modules: module one was a novel transposon and module two comprised a Tn1696-like transposon. The MDR region showed an organization very similar (99% nucleotide sequence identity with a query coverage of 90%) to the plasmid pOZ181 (accession number CP016764), a plasmid from a C. freundii B38 isolate from a hospital in 1998 in China (Appendix A).

The tellurium resistance-associated region located from 8380 bp to 23,064 bp consists of terF-terE-terD-terC-terB-terA-terZ as well as terW-terY-terX. Seven intervening ORFs were identified between terZ and terW. The ter cluster was highly (100%) similar to the ter cluster in p362713-HI3 from K. pneumoniae 362713, the plasmid unnamed1 from K. pneumoniae FDAARGOS_439, p1 from K. oxytoca pKOX3, and p1 from K. pneumoniae 20467, among others. The mercury resistance cluster located from 35,301 bp to 39,277 bp consists of merR-merT-merP-merC-merA-merD-merE. The mer cluster is identical (100%) to those of p1 from *K. pneumoniae* BA2275, pSW37-267106 from S. Worthington OLF-FSR1, the plasmid from *E. coli* S15FP06257, and pMS-37 from *Enterobacter hormaechei* EGYMCRVIM, among others. The silver resistance cluster located from 155,928 bp to 165,278 bp, consisted of silE-silS-silR-silC-silB-silA, which was identical (100%) to those of pLH94-1 from *K. pneumoniae*, the plasmid unnamed1 from *K. pneumoniae* KSB1_7F-sc-2280268, pVNCKp115 from *K. pneumoniae* VNCKp115, and pCAV2018-177 from *K. pneumoniae* CAV2018, among others. The copper resistance-associated region located from 169,677 bp to 175,162 bp consisted of copE-copA-copB-copC-copD-copR. The cop operon was highly (100%) similar to the cop operon in plasmids unnamed1 from *K. pneumoniae* KSB1_7F-sc-2280268, pVNCKp115 from *K. pneumoniae* VNCKp115, pLH94-1 from K. pneumoniae, and the chromosome from *K. pneumoniae* NCTC9180, among others. It has been reported that tellurium, mercury, and copper heavy metal resistance genes have been identified in the IncH plasmid pH11 from a clinical *K. pneumoniae* isolate [28]. A large virulence IncH plasmid, pLVPK, harboring copper, silver, and tellurite resistance genes was previously detected in a bacteremic isolate of K. pneumoniae CG43 [29]. To the best of our knowledge, this is the first report that the IncH plasmid from K. pneumoniae contained these four kinds of heavy metal resistance genes, suggesting that the IncH plasmid from *K. pneumoniae* may be a major reservoir for heavy metal resistance genes. Since heavy metal resistance is not the main focus of this study, we did not investigate the phenotype of the resistance to heavy metals of *K. pneumoniae* 200.

We identified a novel transposon in module one of the MDR regions in pYhe2001, designated Tn6897 in the Tn Number Registry (https://transposon.lstmed.ac.uk/ (accessed on 22 June 2020). Tn6897, flanked by direct copies of intact IS26, contained four direction-similar IS26 and one direction-reverse IS26. IS6 family elements, IS26, have played a pivotal role in the dissemination of resistance determinants in gram-negative bacteria [5,30]. This transposon harbored two transposons, mutated Tn4352 and Tn6020b-1 (Figure 3).

Mutated Tn4352 was bound by an 8-bp repeat region (CATCGGCG) on the right; however, the left target site (GATTGGG) was truncated, and the base was mutated. Mutated Tn4352 comprised two intact IS26 sequences that flanked the kanamycin resistance gene aph(3’)-Ia in Tn6897. Because of Tn4352, tniAΔ1 was interrupted in the downstream, leading to an 8 bp repeat region (CATCGGCG) on the left. The two IS26 flanked tniAΔ1 in different directions. Tn6897 comprised a part of a class 1 integron that included the truncated 5’-CS (including truncated intI1 gene), the gene cassette aadA2, and the qacEΔ1 and sul1 genes in the 3’-CS. It has been reported that truncated class 1 integrons still have the ability to resist to a greater number of antimicrobials in *E. coli* [31]; therefore, the truncated class 1 integron in Tn6897 may confer resistance to antibiotics. The truncated class 1 integron was between the gene for puromycin N-acetyltransferase protein and an IS26. Further upstream included an intact reverse Tn6020b-1 transposon, which included an intact IS26, an IS26Δ1, and aminoglycoside resistance gene aphA1-1. Through BLASTN searches, a portion of Tn6897 (97% nucleotide sequence similarity) was identified in three different strains, including *K. pneumoniae*, *S. enterica*, and *Proteus mirabilis*. Although high nucleotide sequence similarity occurred in chromosomes or plasmids of different strains, no similar structure was found. Tn6897 had two completely similar repeats of IS26 flanked by 14-bp repeats at both ends (Figure 3), indicating mobility potential. Since Tn6897 contained transposons and truncated class 1 integrons, it has the potential to transfer antibiotic resistance genes in different strains.

We found that the tniA gene was interrupted by Tn4352, leading to an 8-bp repeat region (CATCGGCG) downstream. Furthermore, tniAΔ1 was inserted by IS26, leading to a left inverted repeat (IRL) of IS26. In addition, an intact sul1-type class 1 integron was inserted by an IS26 leading to a right inverted repeat (IRR) of IS26. Reverse Tn6020b-1 had a truncated IS26. Furthermore, we failed to observe direct repeats (DRs) flanking IS26 as well as specific target site duplication patterns, suggesting that the Tn6897 conformation may have occurred by IS26-mediated homologous recombination rather than transposition [32]. These observations revealed that Tn6897 could be formed with the help of IS26 and other transposons. Here, we put forward a hypothesis (Figure 4). To explain the evolution of Tn6896, a hypothetical sequence of 4065 bp, comprising a gene encoding N-acetyltransferase and an intact sul1-type class 1 integron, was sent to BLASTn analysis. The analysis identified matches (100% coverage) to those in plasmids from S. Dublin CVM 22429, S. Anatum str. USDA-ARS-USMARC-1736, and S. Dublin 853. Figure 4 depicts a reverse IS26 inserted in the 5’CS of tniA. Additionally, a Tn4352 inserted into the 3’CS of tniA generating an 8-bp repeat region (CATCGGCG). On the other hand, a Tn6023-like including two IS26 in reverse directions and gene aphA1-1 was inserted downstream of the hypothetical sequence, with a reverse IS26 inserted into intI1 of the hypothetical sequence, and another IS26 was inserted in the upstream of the N-acetyltransferase gene. The sequences described above have one copy of IS26 at both ends. Then, two sequences were recombined by recombination between the copies of IS26 in the same orientation. Finally, the sequence above is inserted by IS26 into the downstream IS26, leading to the generation of Tn6897.

Module two was reversed compared with the plasmid pOZ181 from a *C. freundii* B38, which consists of a Tn1696-like transposon (Appendix A). The right IR was interrupted by IS4321, and there was no left IR because of interruption of ISCR1 in the Tn1696-like transposon. Furthermore, a sul1-type class 1 integron that had a 5’CS and four cassettes, aadA5-gcu37-dfrA1-orf, was flanked upstream by a Tn1696-like tnpR-tnpA and downstream by 3’ CS and an ISCR1 transposase. In addition, other resistance genes included the 16S rRNA methylase gene armA downstream of ISCR1. Truncated ISEc29 was located downstream of armA; however, intact ISEc28 was located upstream of armA. Module two showed 99% similarity to the plasmid pBSI034-MCR9 from E. cloacae BSI034, the plasmid pOZ181 from *C. freundii* B38, the plasmid pSIM-1-BJ01 from K. pneumoniae 13624, and the plasmid pWLK-238550 from Raoultella ornithinolytica WLK218, which means that Tn1696-like transposons can be found among different bacteria.

### 2.4. Transformation Experiment

Transformation of plasmids into *E. coli* DH5α was achieved at a frequency of 10^−11^ cells per recipient cell. Genes aadA5 and aac(6’)-Ib-cr were detected in transformant. The MICs of *K. pneumoniae* 200, transformant and DH5α, are shown in Table 1. The MIC of gentamicin for K. pneumoniae 200 was 256 mg/L, while the MIC of gentamicin for E. coli DH5α was 0.125 mg/L. In particular, the MIC of gentamicin for the transformant was higher than that of *E. coli* DH5α, which corresponds to a 256-fold increase in gentamicin MIC because of the presence of plasmids.

## 3. Materials and Methods

### 3.1. Bacterial Strain

The MDR *K. pneumoniae* 200 examined in this study was originally isolated from the liver of a swine suffering from diarrhea at a commercial swine farm in Guangzhou City, Guangdong Province, China, in 2017. *E. coli* DH5α was used as the recipient strain in transformation experiments.

### 3.2. Antimicrobial Susceptibility Testing

Antibiotic susceptibility was determined using the disk diffusion method [11] following the guidelines of the Clinical Laboratory Standards Institute [33]. The disks (OXOID, Hampshire, UK) used in the test were cefotaxime (30 µg), ceftazidime (30 µg), cefoxitin (30 µg), oxacillin (50 µg), meropenem (10 µg), imipenem (10 µg), amikacin (30 µg), gentamicin (10 µg), kanamycin (30 µg), streptomycin (10 µg), ciprofloxacin (5 µg), chloramphenicol (30 µg), erythromycin (15 µg), tetracycline(30 µg), and trimethoprim-sulfamethoxazole (1.25/23.75 µg). *E. coli* ATCC 25922 was used for quality control.

### 3.3. Genome Sequencing, Genome Assembly and Bioinformatics

To comprehensively understand the genetic basis of the resistance of *K. pneumoniae* 200, the complete genome sequence was generated by WGS using PacBio RSII (Pacific Biosciences, Menlo Park, CA, USA) and Illumina MiSeq (Illumina, San Diego, CA, USA) platforms as previously described [34]. WGS data were assembled using SOAPdenovo v1.05 software. Circularization was achieved by manual comparison and removal of a region of overlap, and the final genome was confirmed by remapping the sequence data. The assemblies yielded a circular chromosome and two circular plasmids. Gene prediction was performed using GeneMarkS, and whole-genome BLAST searches (E-value ≤ 1 × 10^−5^, minimal alignment length percentage ≥ 80%) were performed against 5 databases: Kyoto Encyclopedia of Genes and Genomes (KEGG), Clusters of Orthologous Groups (COG), NCBI nonredundant protein database (NR), Swiss-Prot, and Gene Ontology (GO). Plasmid incompatibility groups were identified using the online database PlasmidFinder (https://cge.cbs.dtu.dk/services/PlasmidFinder/ (accessed on 2 February 2020). Antimicrobial resistance genes were identified using the ResFinder 3.1 tool (https://cge.cbs.dtu.dk/services/ResFinder/ (accessed on 2 February 2020) with an identity threshold of 96%. Integrons were analyzed using the integron identification tool INTEGRALL (http://integrall.bio.ua.pt/ (accessed on 2 February 2020). *K. pneumoniae* virulence genes were identified with the aid of the *K. pneumoniae* section of the Institut Pasteur MLST and whole genome MLST databases (http://bigsdb.Pasteur.fr (accessed on 2 February 2020). For sequence comparisons, the BLAST algorithm (www.ncbi.nlm.nih.gov/BLAST (accessed on 2 February 2020) was used. Multilocus sequence typing (MLST) analysis of *K. pneumoniae* 200 and cgMLST phylogenetic relationship analyses of all public genomesequences of ST37 were performed using the BacWGSTdb server with a threshold of 1000. The DNA sequences of chromosomes and plasmids of *K. pneumoniae* 200 were deposited in NCBI GenBank with the accession numbers CP055293, CP062278, and CP063211, respectively.

### 3.4. Transformation Experiments

Transformation of plasmid DNA isolated from *K. pneumoniae* 200 into *E. coli* DH5α was performed as described previously [35]. *K. pneumoniae* 200 and DH5α strains were first grown in 25 mL of LB liquid medium overnight at 37 °C with shaking. Plasmid DNA was extracted from *K. pneumoniae* 200 with a Plasmid Mini Purification Kit (Amersham Biosciences, Uppsala, Sweden). Then, a 1-mL culture of *E. coli* DH5α was diluted in 100 mL of LB liquid medium and incubated at 37 °C until the cells reached the early exponential growth phase. Cells were then centrifuged twice within cooled 0.1 M CaCl_2_ for 10 min at 2700× *g* and finally resuspended in 500 µL of LB liquid medium. All centrifugation steps were performed at room temperature (RT) at approximately 25 °C. Four aliquots of 100 µL of cells were prepared, and plasmidic DNA were added and gently mixed, respectively. The remaining 100-µL cell aliquot was used as a negative control (no DNA was added). Samples were incubated at 37 °C for 1 h. After that, three aliquots were plated on LB plates supplemented with gentamicin (32 µg/mL), while the fourth aliquot was serially diluted and plated on LB plates without antibiotic to enumerate recipient cells. Negative control was also plated on LB plates supplemented with gentamicin (32 µg/mL). All the plates were incubated at 37 °C overnight. Transformation efficiency was calculated based on the ratio of transformants to the total number of viable cells. Experiments were performed with three biological replicates. To identify the *aadA5* gene in transformants, primers 5’-CGCTCAACGCAAGATTCTCT-3’ (forward), and 5’-ATGGGTGAATTTTTCCCTGCAC-3’ (reverse) for *aadA5* (792 bp) were used in PCR. Furthermore, the presence of the *aac(6’)-Ib-cr* gene in transformants was confirmed by PCR amplification followed by DNA sequence analysis. Primers for *aac(6**’)-Ib-cr* (612 bp) were 5’-AAGGGTTAGGCATCACTGCG-3’ (forward) and 5’-AGACATCATGAGCAACGCAA-3’ (reverse). The primers were designed using NCBI Primer-BLAST. PCR conditions were: initial denaturation at 95 °C for 5 min, 30 cycles of amplification (30 s at 95 °C, 30 s at 55 °C, and 90 s at 72 °C), followed by an extension at 72 °C for 10 min. PCR products were purified and sequenced by Majorbio Company (Shanghai, China). MICs of *E. coli* DH5α and five transformants were determined by Etest (Liofilchem S.R.L.) according to the manufacturer’s instructions. *E. coli* ATCC 25922 served as a quality control strain.

## 4. Conclusions

To the best of our knowledge, we described a novel plasmid pYhe2001 in a *K. pneumoniae* isolate of swine origin for the first time. A novel Tn*6897* that was identified in the plasmid pYhe2001 likely underwent a recombination event and showed a high potential for resistance development. Heavy metal resistance genes were identified in the plasmid pYhe2001, which expanded the spectrum of IncH plasmids. This is the first time that the IncF plasmid carrying *bla*_CTX-M-27_ was discovered in the liver of a swine, which warrants investigation of the prevalence of *bla*_CTX-M-57_ harboring *K. pneumoniae*. The results of this study provide additional evidence of the variation in MDR *K. pneumoniae* that threatens the health of humans and animals.

## Figures and Tables

**Figure 1 antibiotics-11-01225-f001:**
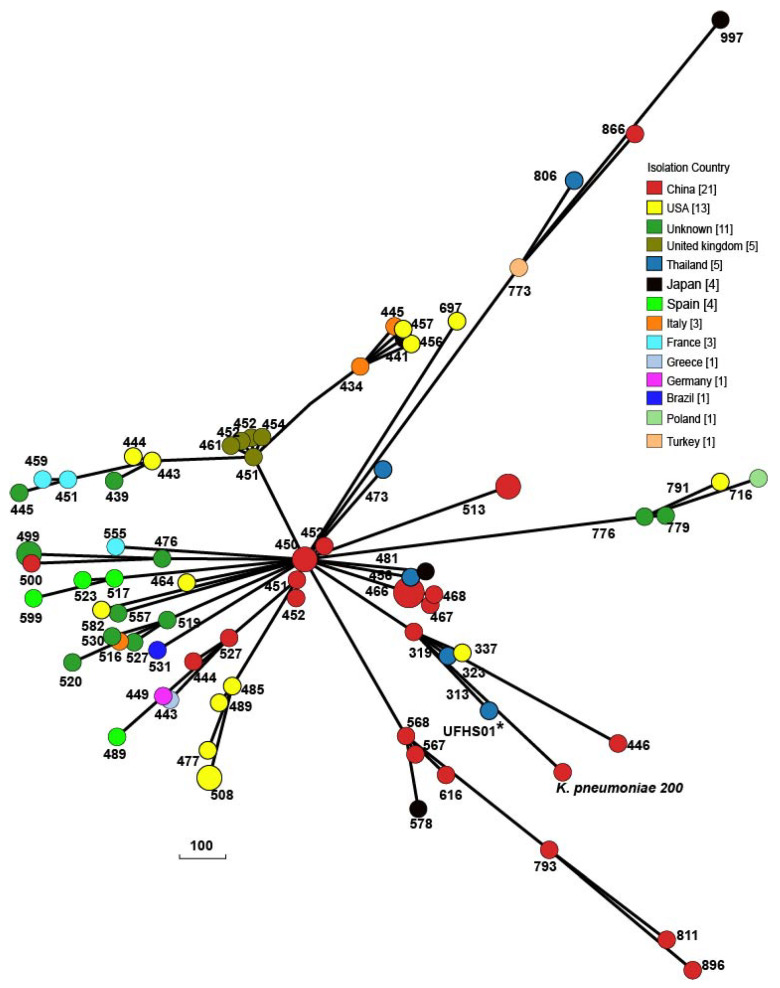
Phylogenetic trees of all ST37 K. pneumoniae strains of released public sequences based on cgMLST. The evolutionary distance showed that *K. pneumoniae* 200 was related to human-associated *K. pneumoniae* 4300STDY6470454 (UFHS01) from Thailand. The numbers indicate the number of allele differences between *K. pneumoniae* 200 and other isolates.

**Figure 2 antibiotics-11-01225-f002:**
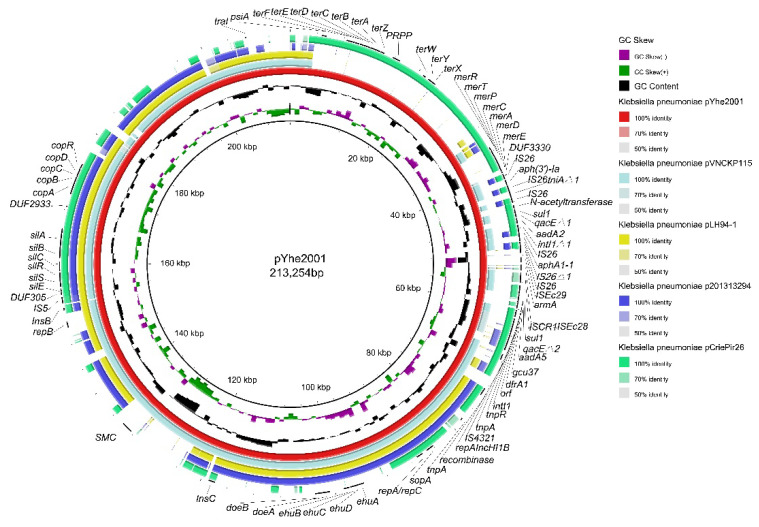
Comparisons of pYhe2001 complete genome sequences with NCBI-published complete plasmid sequences of another four *K. pneumoniae* strains using BRIG. The innermost circles represent the GC skew (purple/green) and GC content (black). Rings 1–5 represent *K. pneumoniae* pYhe2001, *K. pneumoniae* pVNCKp115, *K. pneumoniae* pLH94-1, *K. pneumoniae* p201313294, *K. pneumoniae* pCriePir26, respectively.

**Figure 3 antibiotics-11-01225-f003:**
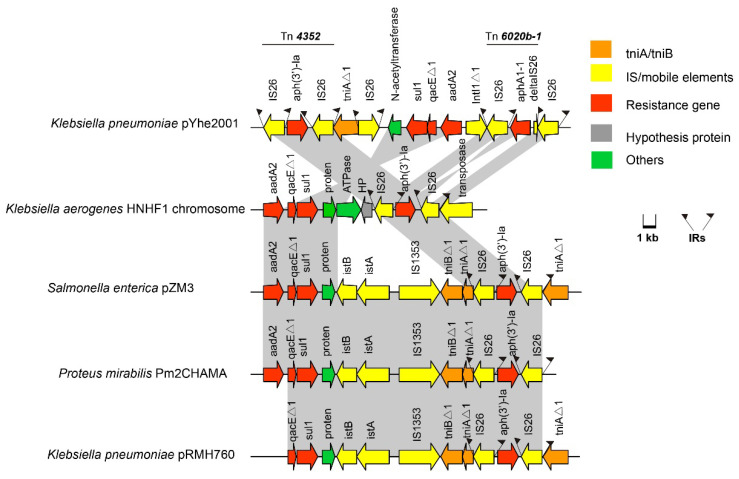
Organization of the Tn6897 transposon in the plasmid pYhe2001 and comparison with a similar structure. ORFs are shown as arrows, indicating the transcription direction, and the colors of the arrows represent different fragments. Intact ISs are represented by arrows, showing the direction of transcription of the transposase genes. Flags represent the IRs of ISs and transposons. Homologous gene clusters in different isolates are shaded in gray (>97%).

**Figure 4 antibiotics-11-01225-f004:**
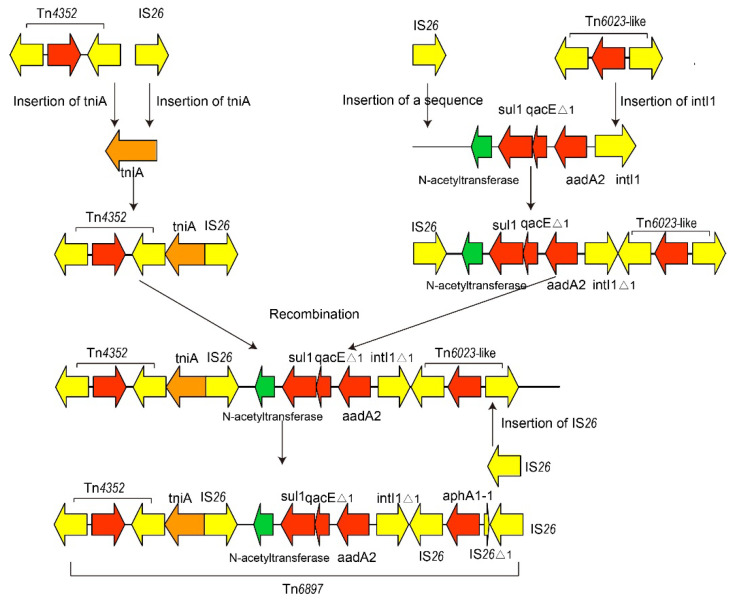
Genealogy of Tn6897 in plasmid pYhe2001. Due to the lack of DRs flanking IS26 as well as specific target site duplication patterns, the novel Tn6897 could be hypothesized to produce by IS26 recombination and other transposons.

**Table 1 antibiotics-11-01225-t001:** MICs * of *K. pneumonia* and transformant.

MIC (mg/L)	*K. pneumonia* 200	Transformant	DH5α
Amikacin	>256	128	1
Ampicillin	>256	>256	2
Ciprofloxacin	>256	32	0.008
Chloramphenicol	>256	32	2
Gentamicin	256	32	0.125
Kanamycin	>256	128	0.75
Oxacillin	>256	64	0.25
Streptomycin	64	16	1

* minimum inhibitory concentration.

## Data Availability

The chromosomal DNA and plasmids of *K. pneumoniae* 200 were deposited in NCBI GenBank with the accession numbers CP055293, CP062278, and CP063211, respectively.

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
