# Peer review of "Identification of a Novel IncHI1B Plasmid in MDR Klebsiella pneumoniae 200 from Swine in China"

_antibiotics, 2022, doi:10.3390/antibiotics11091225_

Round 1
Reviewer 1 Report
The current manuscript provides the whole-genome sequencing of K. pneumonia 200 isolated from infected swine. This work is descriptive, I have nothing to add.
Author Response
Reviewer 1:
The current manuscript provides the whole-genome sequencing of K. pneumonia 200 isolated from infected swine. This work is descriptive, I have nothing to add.
Response: Thanks for the comments.
Reviewer 2 Report
Manuscript Huixian Liang et al. studies multidrug-resistant Klebsiella pneumoniae strain 200. Despite the fact that the work is not innovative and mainly contains already known information, it is necessary to identify and describe as many new plasmids as possible that can serve as a mobile reservoir for virulence genes. The authors focused on the characterization not only of the newly isolated plasmids, but also of the genome itself. It follows that the analysis of genome and plasmid sequences is at a good level, but it was mainly devoted to mobile elements and their genealogy (Tn6897 in pYhe2001). However, the authors also described several facts with the potential for further investigation. For example, they paid little attention to resistance to heavy metals. Nevertheless, I consider the work can be valuable and publishable. I have several comments on the work:
1. I recommend to correct the writing style, for example, on the prepositions used...
2. Complete the missing citations in the manuscript ( lines 115-120)
3. In the work that you present as the first record of the occurrence of blaCTX-M-27 in the IncF plasmid (pYhe2002), you did not describe the allele (introduction or discusson)
4. Include Table S5 directly in the manuscript and determine the expression level of resistance genes in your strain and in E.coli DH5α
5. Determine the sensitivity of the K. pneumoniae strain 200 to heavy metals, especially tellurium. The sensitivity can be determined relatively quickly and easily (Chaturvedi et al. 2015 (DOI 10.1007/s00203-015-1147-7)
Author Response
Reviewer 2:
Manuscript Huixian Liang et al. studies multidrug-resistant Klebsiella pneumoniae strain 200. Despite the fact that the work is not innovative and mainly contains already known information, it is necessary to identify and describe as many new plasmids as possible that can serve as a mobile reservoir for virulence genes. The authors focused on the characterization not only of the newly isolated plasmids, but also of the genome itself. It follows that the analysis of genome and plasmid sequences is at a good level, but it was mainly devoted to mobile elements and their genealogy (Tn6897 in pYhe2001). However, the authors also described several facts with the potential for further investigation. For example, they paid little attention to resistance to heavy metals. Nevertheless, I consider the work can be valuable and publishable. I have several comments on the work:
- I recommend to correct the writing style, for example, on the prepositions used....
Response: Thanks for the comments and suggestions. we have sent the manuscript for English editing. For example, prepositions used in manuscript were corrected as follows:
“by natural transformation” changed into “through transformation together” (line 21)
“transport for multiple resistance genes” changed into “transport multiple resistance genes” (line 23)
“As an increasing number of antibiotics” changed into “With an increasing number of antibiotics” (line 31)
“on the chromosome” changed into “in the chromosome” (line 74)
“on the chromosome” changed into “in the chromosome” (line 75)
“compared with strains HS11286 and ZYST1” changed into “compared to strains HS11286 and ZYST1” (line 93)
“the potential ability to uptake genes” changed into “the potential ability of uptaking genes” (lines 95-96)
“detected from IncF plasmids” changed into “detected in IncF plasmids” (line 133)
“compared to pYhe2001” changed into “compared with pYhe2001” (lines 153-154)
“compared to pVNCKp115” changed into “compared with pVNCKp115” (line 163)
“compared to the plasmid pOZ181” changed into “compared with the plasmid pOZ181” (line 262)
- Complete the missing citations in the manuscript (lines 115-120)
Response: We have added citations in the manuscript (lines 117-124).
- In the work that you present as the first record of the occurrence of blaCTX-M-27in the IncF plasmid (pYhe2002), you did not describe the allele (introduction or discussion)
Response: We have added the allele description in discussion (lines 130-136) as follows:
The expression of CTX-M β-lactamases genes were found commonly in K. pneumoniae strains all over the world. Many kinds of CTX-M allele were found, such as blaCTX-M-55 [21], blaCTX-M-63 [22], and blaCTX-M-15 [23-24]. The predominant ESBL allele blaCTX-M-15 was commonly detected in IncF plasmids from K. pneumoniae [23-24]. From the results of PLSDB, among the 681 hits , gene blaCTX-M-27 could be found in plasmids carried by K. pneumoniae from different sources but not liver source (Table S3). This is the first report that an IncFIA plasmid with the blaCTX-M-27 gene from swine liver K. pneumoniae isolate.
Reference:
Stosic MS, Leangapichart T, Lunha K, Jiwakanon J, Angkititrakul S, Järhult JD, Magnusson U, Sunde M. Novel mcr-3.40 variant co-located with mcr-2.3 and blaCTX-M-63 on an IncHI1B/IncFIB plasmid found in Klebsiella pneumoniae from a healthy carrier in Thailand. J Antimicrob Chemother. 2021 76(8):2218-2220. DOI: 10.1093/jac/dkab147
Cao X, Zhong Q, Guo Y, Hang Y, Chen Y, Fang X, Xiao Y, Zhu H, Luo H, Yu F, Hu L. Emergence of the Coexistence of mcr-1, blaNDM-5, and blaCTX-M-55 in Klebsiella pneumoniae ST485 Clinical Isolates in China. Infect Drug Resist. 2021 Aug 28;14:3449-3458. DOI: 10.2147/IDR.S311808
Gancz A, Kondratyeva K, Cohen-Eli D, Navon-Venezia S. Genomics and virulence of Klebsiella pneumoniae Kpnu95 ST1412 harboring a novel Incf plasmid encoding blactx-M-15 and qnrs1 causing community urinary tract infection. Microorganisms. 2021 9(5):1022. DOI: 10.3390/microorganisms9051022
Mshana SE, Hain T, Domann E, Lyamuya EF, Chakraborty T, Imirzalioglu C. Predominance of Klebsiella pneumoniae ST14 carrying CTX-M-15 causing neonatal sepsis in Tanzania. BMC Infect Dis. 2013 13:466. DOI: 10.1186/1471-2334-13-466
- Include Table S5 directly in the manuscript and determine the expression level of resistance genes in your strain and incoli DH5α
Response: We have changed Table S5 as Table 1 in the manuscript. We have added “The MIC of gentamicin for K. pneumoniae 200 was 256 mg/L. While, The MIC of gentamicin for E. coli DH5α was 0.125 mg/L.” (lines 277-279) to describe the expression level of resistance genes in my strain and in E.coli DH5α.
- Determine the sensitivity of the pneumoniaestrain 200 to heavy metals, especially tellurium. The sensitivity can be determined relatively quickly and easily (Chaturvedi et al. 2015 (DOI 10.1007/s00203-015-1147-7)
Response: Thank you for the suggestion. In this manuscript, the aim of our research was to characterize a novel IncHI1B plasmid in MDR Klebsiella pneumoniae 200 via genome comparison and genes analysis. We understand that it would provide additional information to determine the sensitivity of the K. pneumoniae strain 200 to heavy metals. However, the focus on this study was antibiotic resistance instead of resistance to heavy metals. In addition, high purity reagents needed be purchased foreign countries (materials: silver nitrate, copper sulfate, sodium tellurite; Supplier: Sigma-Aldrich) for related tests. Due to the pandemic, there is an extremely long wait time to obtain those materials. Therefore, we would focus on the sensitivity of the K. pneumoniae strain 200 to heavy metals to in the next stage.
Reviewer 3 Report
This manuscript aims to investigate the genome of a MDR strain of Klebsiella pneumoniae, isolated from a swine. In the main focus of the study were mobile genetic elements, i.e. plasmids and transposons. I went through the current manuscript with great interest and got the following impression.
1) There is an inconsistency in plasmid designation in the title/abstract and conclusion: IncHI1B or pYhe2001 was the novel plasmid, which you discovered?
2) In the text of the paper I could not find direct evidence for the last sentence of the Abstract: This study first reported a novel IncHI1B plasmid from swine origin K. pneumoniae 200, suggesting that the plasmids may act as reservoirs for various antimicrobial resistance genes and transport for multiple resistance genes in K. pneumoniae of both animal and human origin.
3) The manuscript is not optimal regarding the quality of the English, and that effort is necessary to make the text more intelligible.
4) Please, carefully check that you use italic type for bacterial taxa in the whole paper and that you follow the rules of microbial nomenclature: lines 91-95, 109, etc.
Overall, I conclude that the paper may be accepted after minor revision. I believe that the reviewed paper is of interest to the readers of Antibiotics.
Author Response
Reviewer 3:
This manuscript aims to investigate the genome of a MDR strain of Klebsiella pneumoniae, isolated from a swine. In the main focus of the study were mobile genetic elements, i.e. plasmids and transposons. I went through the current manuscript with great interest and got the following impression.
1)There is an inconsistency in plasmid designation in the title/abstract and conclusion: IncHI1B or pYhe2001 was the novel plasmid, which you discovered?
Response: Thanks for the comments and suggestions. In our manuscript, the name of a novel plasmid was pYhe2001. IncHI1B was its type description, not its name.
2) In the text of the paper I could not find direct evidence for the last sentence of the Abstract: This study first reported a novel IncHI1B plasmid from swine origin K. pneumoniae 200, suggesting that the plasmids may act as reservoirs for various antimicrobial resistance genes and transport for multiple resistance genes in K. pneumoniae of both animal and human origin.
Response: Thanks for the suggestions. To support the conclusion, we have added “The plasmids included in the hits were mostly from K. pneumoniae and the remaining plasmids were from other Enterobacteriaceae species (e.g.,Escherichia coli, Salmonella enterica and Citrobacter freundii). These plasmids were collected from humans (e.g., clinical patients), food (e.g., pork and milk), the environment (e.g., rivers and air) and animals (e.g., chickens and swine)” in lines 145-149.
3) The manuscript is not optimal regarding the quality of the English, and that effort is necessary to make the text more intelligible.
Response: We have sent the manuscript for English editing, and it has made more intelligible thoroughly.
4) Please, carefully check that you use italic type for bacterial taxa in the whole paper and that you follow the rules of microbial nomenclature: lines 91-95, 109, etc.
Response: We have checked and correct the italic type for bacterial taxa in the whole paper as follows:
“K. pneumoniae 200” changed into “K. pneumoniae 200” (line 95)
“ST37 K. pneumoniae” changed into “ST37 K. pneumoniae” (line 98)
“C. freundii B38” changed into “C. freundii B38” (line 171)
“Klebsiella pneumoniae” changed into “Klebsiella pneumoniae” (line 409)
“Klebsiella pneumoniae” changed into “Klebsiella pneumoniae” (line 411)
“Klebsiella pneumoniae” changed into “Klebsiella pneumoniae” (line 452)
Overall, I conclude that the paper may be accepted after minor revision. I believe that the reviewed paper is of interest to the readers of Antibiotics.